# Properties of a Dental Adhesive Containing Graphene and DOPA-Modified Graphene

**DOI:** 10.3390/polym16142081

**Published:** 2024-07-21

**Authors:** Renata Pereira, Rodrigo Barros Esteves Lins, Elton Faria de Souza Lima, Maria do Carmo Aguiar Jordão Mainardi, Stephani Stamboroski, Klaus Rischka, Flávio Henrique Baggio Aguiar

**Affiliations:** 1Department of Restorative Dentistry, Division of Operative Dentistry, Piracicaba Dental School, University of Campinas (UNICAMP), Av. Limeira 901, Piracicaba 13414-903, SP, Brazil; re_pe@hotmail.com (R.P.); camoajm@hotmail.com (M.d.C.A.J.M.); baguiar@unicamp.br (F.H.B.A.); 2Department of Adhesive Bonding Technology and Surfaces, Fraunhofer Institute for Manufacturing Technology and Advanced Materials (IFAM), Wiener Straße 12, 28359 Bremen, Germany; stephani.stamboroski@ifam.fraunhofer.de; 3School of Dentistry, Federal University of Alagoas, Lourival Melo Mota Ave, Maceió 57072-900, AL, Brazil; rodrigo.lins@foufal.ufal.br; 4Federal Institute of Education, Science and Technology of Goiás (IFG—Campus Uruaçu), Rua Formosa, Qd 28 e 29—Loteamento Santana, Uruaçu 76400-000, GO, Brazil; elton.lima@ifg.edu.br

**Keywords:** graphene, L-DOPA, dental adhesive, anti-bacterial agents, dental materials

## Abstract

Graphene is a promising biomaterial. However, its dispersion in aqueous medium is challenging. This study aimed to modify graphene nanoparticles with L-dopa to improve the properties of experimental dental adhesives. Adhesives were formulated with 0% (control), 0.25%, 0.5%, and 0.75% of graphene, modified or not. Particle modification and dispersion were microscopically assessed. Degree of conversion was tested by Fourier-transform infrared spectroscopy. Flexural strength and modulus of elasticity were evaluated by a 3-point flexural test. Bond strength was tested by shear. To test water sorption/solubility, samples were weighed during hydration and dehydration. Antibacterial activity was tested by *Streptococcus mutans* colony-forming units quantification. Cytotoxicity on fibroblasts was evaluated through a dentin barrier test. The modification of graphene improved the particle dispersion. Control presented the highest degree of conversion, flexural strength, and bond strength. In degree of conversion, 0.25% of groups were similar to control. In bond strength, groups of graphene modified by L-dopa were similar to Control. The modulus of elasticity was similar between groups. Cytotoxicity and water sorption/solubility decreased as particles increased. Compared to graphene, less graphene modified by L-dopa was needed to promote antibacterial activity. By modifying graphene with L-dopa, the properties of graphene and, therefore, the adhesives incorporated by it were enhanced.

## 1. Introduction

Secondary caries formation has been widely considered one of the most important and common reasons for restoration failures, regardless of the restorative material type [1,2,3]. Under the assumption that secondary caries occurs next to preexisting restorations [1,4,5], one of the main strategies to prevent such lesions is by incorporating antimicrobials or remineralizing agents into restorative materials [6,7]. Adhesive systems with antibacterial activity, specifically, have been extensively developed in an effort to obtain, at the same time, effective reduction of dental pathogens, durability of antibacterial action, maintenance of adhesive system properties, and biocompatibility with dental tissues [8].

Graphene is a carbon-based material that has attracted widespread attention, since Novoselov et al. isolated it from graphite through mechanical exfoliation and reported its unique electronic properties [9]. Due to its aromatic, two-dimensional conjugated structure of carbon atoms, graphene exhibits remarkable electronic, thermal, optical, mechanical, and biological-related properties [10]. Among its main characteristics, high mechanical strength, high modulus of elasticity, high flexibility, high electron mobility at room temperature, high thermal conductivity, biocompatibility, and antimicrobial protection are noticed [10,11,12,13].

Several mechanisms are involved in the antimicrobial action of graphene: cell membrane damage, destructive extraction of phospholipids from lipid membranes, oxidative stress and separation of microorganisms from their microenvironment [13,14,15]. Yet, in the restorative dentistry field, reports of the use of graphene as an antimicrobial agent are relatively scarce. In a commercial dental adhesive, in particular, solely Bregnocchi and colleagues incorporated graphene in order to test its antibacterial activity [16]. Indeed, decreased viability of *Streptococcus mutans* was reported. However, the adhesive with the highest concentration of graphene (0.5 wt%) generated a high rate of premature failure [16]. In this context, one should consider that, although graphene-based materials are promising, the controllable dispersion of the graphene particles within composites is extremely challenging due to their hydrophobic nature, lower surface energy, and high van der Waals force between their layers, which lead to agglomerations [17,18,19].

One strategy to stabilize the dispersion of graphene is by modifying it with the addition of amphiphilic compounds, which reduce the solvent’s surface energy, promote exfoliation, and prevent re-aggregations [19]. Amino acids with catechol functional residues (moieties), such as 3,4-dihydroxyphenyl-L-alanine (L-DOPA), might work as an amphiphilic dispersant in this case [20]. Catechol moieties are used by mussels as one of the key functional groups for adhesion to a variety of wet surfaces [21]. It has been reported that these compounds may act as antioxidants and chelating agents, interacting strongly with organic or inorganic substrates [21,22,23]. Yet, they are able to promote nanoparticle stabilization [20,24]. In the restorative dentistry field, specifically, it was demonstrated that catechol-functionalized polymers might bind to dentin surfaces [25] and improve the properties of hybrid layer-adhesive interfaces [23].

Taking into account all features presented by the catechol moieties, the aim of this study was to evaluate the particle modification and dispersion, degree of conversion, flexural strength, modulus of elasticity, bond strength, fracture pattern, water sorption and solubility, antibacterial activity, and cytotoxicity of an experimental adhesive incorporated by graphene and graphene modified by L-DOPA.

## 2. Materials and Methods

### 2.1. Modification of Graphene by L-DOPA and Analysis of Particles Modification/Dispersion

A solution of 5 mL of phosphate-buffered saline (PBS) (pH 6.0), 100 mg of graphene nanoplatelet aggregates (sub-micronparticles, 500 m^2^/g surface area), 5 mg of laccase from *Trametes versicolor*, and 5 mg of L-DOPA was formulated and stirred on a mixer for 20 h. Graphene was supplied by Alfa Aesar (Thermo Fisher Scientific, Kandel, Germany), while laccase and L-DOPA were respectively purchased from ASA Spezialenzyme GmbH (Wolfenbüttel, Germany) and Sigma-Aldrich Chemie GmbH (Steinheim, Germany).

The solution was centrifuged three times (Universal centrifuge Heraeus Megafuge 16R, Thermo Fisher Scientific, Osterode am Harz, Germany) and rinsed successively with deionized water. Finally, the pellet obtained was dried by lyophilization (Freeze dryer Loc-1m Alpha 1–4, Martin Christ Gefriertrocknungsanlagen GmbH, Osterode am Harz, Germany).

Laccase was included in the formulation considering the polymerization of L-DOPA would be faster and more effective, as shown by a previous study [25].

The particle modification was assessed by scanning electron microscopy (SEM). A total of 0.5 mg of particles of graphene (G) and graphene modified by L-DOPA (Gd) were sputter-coated with 7 nm of gold using an EM ACE600 high vacuum sputter coater (Leica Microsystems, Wetzlar, Germany). Image magnifications of 1000× and 5000× of the particle surfaces were carried out using a desktop SEM (Phenom XL, Phenom-World BV, Eindhoven, The Netherlands) at an acceleration voltage of 10 kV. The dispersion of 2 mg/mL of G and Gd in water and ethanol was visually analyzed and compared after 24 h.

### 2.2. Formulation of Dental Adhesives

An experimental light-curing two-step dental adhesive (E) was formulated with the following components, duly weighed in an analytical balance: 7 wt% water; 30 wt% ethanol; 20 wt% bisphenol A glycidyl methacrylate (Bis-GMA); 5 wt% diurethane dimethacrylate (UDMA); 15 wt% 2-hydroxyethyl methacrylate (HEMA); 10 wt% glycerol 1,3-dimethacrylate; 10 wt% polyacrylic acid; 1 wt% ethyl-4-dimethylaminobenzoate (EDAB); 1.5 wt% diphenyliodonium hexafluorophosphate (DPIHP); and 0.5 wt% camphorquinone. The mixture was taken to a speed mixer (DAC 150 FVZ, Hauschild GmbH & Co., KG, Hamm, Germany) for 5 min at 2700 rpm.

The adhesive received the addition of three different concentrations of G or Gd and was mixed again in the speed mixer, generating seven experimental groups:E without addition of G/Gd—control;E with 0.25 wt% G—0.25% G;E with 0.5 wt% G—0.5% G;E with 0.75 wt% G—0.75% G;E with 0.25 wt% Gd—0.25% Gd;E with 0.5 wt% Gd—0.5% Gd;E with 0.75 wt% Gd—0.75% Gd.

Groups 2 to 7 had the percentage of ethanol deducted according to the percentage of G and Gd added.

### 2.3. Microscopic Analysis of Particles Dispersion in the Adhesives

The dispersion of G and Gd particles in the adhesives was analyzed under an optical microscope (Axiostar plus, Carl Zeiss Microscopy GmbH, Jena, Germany). Therefore, 20 µL of each adhesive were stirred, applied to a glass film, and light cured in standard mode: 1000 mW/cm^2^ (Valo, Ultradent Products, South Jordan, UT, USA). The light optical power (mW) delivered by the curing unit was previously confirmed with a power meter (Ophir Optronics, Jerusalem, Israel). The light curing time was standardized to 1 min in the whole study.

### 2.4. Degree of Conversion Analysis

For the degree of conversion analysis, a total of 35 disc-shaped samples (n = 5) were confectioned using a mold of heavy and light-bodied polyvinyl siloxane material (Scan Putty and Scan Light, Yller Biomateriais S/A, Pelotas, RS, Brazil), with an orifice of 5 mm ø × 0.5 mm height. Thirty microliters of each adhesive were dispensed in the orifice, and a polyester strip was placed over the mold. The tip of a light curing unit (Valo, Ultradent Products) was positioned directly over the polyester strip, and the samples were light cured.

The degree of conversion of samples bottom surfaces was analyzed by Fourier transform infrared spectroscopy (FTIR) coupled with an attenuated total reflectance (ATR) device (IRAffinity-1, Shimadzu Corporation, Tokyo, Japan) before and after light curing [26]. Absorption spectra were obtained in a range of 4000–650 cm^−1^, using 64 scans at 4 cm^−1^ resolution and 2.8 mm/s. The height of the absorbance peak was determined after the subtraction of the baseline and normalization process [26] using Origin software (https://www.originlab.com/index.aspx?go=PRODUCTS/Origin, accessed on 23 May 2024, Originlab Corporation, Northampton, MA, USA). The peak band located at 1608 cm^−1^ (aromatic component group) was used as an internal standard [27]. The degree of conversion was calculated through the formula: DC (%) = 100 × [1 − (R cured/R uncured)], where R represents the ratio between aliphatic band absorption at 1638 cm^−1^ and aromatic band absorption at 1608 cm^−1^, respectively. 

### 2.5. Flexural Strength and Modulus of Elasticity Analyses

Except for sample dimensions, ISO 4049 [28] was used to guide the methodology of flexural strength, as well as water sorption and solubility analyses. A total of 35 bar-shaped samples (n = 5) (20 length × 1 height × 2 width mm^3^) were confectioned using a mold of heavy and light-bodied polyvinyl siloxane material (Scan Putty and Scan Light, Yller Biomateriais S/A). The adhesives were applied in the mold until complete filling and were covered by a polyester strip. Each sample was light cured (Valo, Ultradent Products) on top and bottom surfaces, on both extremities, and in the middle. The samples were stored in distilled water at 37 °C for 24 h.

Samples were tested through a 3-point flexural test using a universal mechanical testing machine (Instron 4411, Instron Inc., Canton, MA, USA) at 1 mm/min speed with a 50 N load. The flexural strength (FS), expressed in MPa, was calculated through the formula: FS = 3 FL/2bh^2^, where F is the force required to cause the sample failure, in Newton, L is the distance between the supports, b is the sample width, and h is the sample thickness. The modulus of elasticity (ME), expressed in MPa, was determined considering the sample’s dimension and the slope of the linear portion of the load–deflection curve for each sample tested for flexural strength. Therefore, the following formula was applied: ME = FL^3^/4bh^3^d, where d is the load deflection, in millimeters.

### 2.6. Shear Bond Strength Analysis

A total of 70 freshly extracted, healthy, and cracked-free bovine incisors were selected. The cleaning of outer surfaces was performed through root scalling, followed by pumice prophylaxis application. After cleaning, the teeth crowns had their middle third sectioned with a diamond blade (Isomet Diamond Wafering Blades, Buehler, Lake Bluff, IL, USA) in a metallographic precision cutter (Isomet 1000, Buehler) until they reached 70 blocks of 4 mm^2^. The dental blocks were mounted in epoxy resin (Epodex Ultra Clear, Epodex GmbH, Moers, Germany), and their top surfaces were abraded in a polishing machine (Planopol, Struers, Ballerup, Denmark) with 120-grit silicon carbide abrasive papers under water cooling and constant speed until dentin exposure.

For the restorative procedure, the dentin surface was first conditioned with 35% phosphoric acid (Ultra-Echt, Ultradent Products) for 15 s and rinsed at the same time with demineralized water. A layer of adhesive, according to each experimental group, was applied to the slightly moist dentin surface, with the help of a disposable microbrush in a light scrubbing motion. The adhesive was air dried for 5 s to allow the evaporation of solvents and was then light cured: 1200 mW/cm^2^ ± 10% (Bluephase Style, Ivoclar Vivadent, Schaan, Liechtenstein). With the aid of a bonding clamp fitted with a bonding mold insert (Ultradent Products), a pillar (2.32 mm ø × 2 mm height) of resin composite (Filtek Supreme Flowable Restorative, 3M Oral Care, St. Paul, MN, USA), of A2 shade, was built on the dentin surface. The composite was also light cured for 1 min (Bluephase Style, Ivoclar Vivadent). The teeth were stored under relative humidity at 37 °C for 24 h.

Ten dentin blocks from each group were used for the shear test. The sample was fixed in a standardized device for shear testing (test base clamp, Ultradent Products) in order for the composite pillar to be placed perpendicularly to the shear loading. The device with the sample was positioned in a universal testing machine (Z2.5 TN, Zwick/Roell, Ulm, Germany), and the test was conducted with a 1 kN load cell at 1 mm/min until failure. The adhesive interface area on the side of the failure was measured with an electronic digital caliper (Promat, Dortmund, Germany). The bond strength (BS), expressed in MPa, was calculated through the formula: BS = F/A, where F is the force required to cause the sample failure, in Newton, and A is the cross-sectional failure area, in mm^2^.

### 2.7. Fracture Pattern Analysis

The fracture patterns were evaluated both through visual analysis and by optical microscopy (Axio Vert.A1, Carl Zeiss Microscopy GmbH) and classified as cohesive in dentin, cohesive in composite, or adhesive. Adhesive was specifically classified as cohesive in adhesive; adhesive between adhesive–dentin; adhesive between adhesive–composite; mixed adhesive 1—involving cohesive failure in adhesive and another adhesive pattern; and mixed adhesive 2—not involving cohesive failure in adhesive. Mixed adhesive fractures were defined as having more than one adhesive pattern and none prevailing. The patterns were quantified and converted to percentages.

### 2.8. Water Sorption and Solubility Analyses

A total of 70 disc-shaped samples were confectioned (n = 10) with a light-bodied polyvinyl siloxane mold (Honigum-Mono, DMG Chemisch-Pharmazeutische Fabrik GmbH, Hamburg, Germany), presenting an orifice of 5 mm ø × 1 mm height. Thirty microliters of each adhesive were dispensed in the orifice, and the samples were light cured (Bluephase Style, Ivoclar Vivadent) on the top and bottom surfaces.

Immediately after light curing, samples were stored in a desiccator containing silica gel. They were weighed every 24 h on a calibrated analytical balance until the mass varied less than 0.2 mg (m1). The thickness and diameter of the samples were measured with an electronic digital caliper (Promat) to determine their volume (V) in mm^3^. The samples were then placed individually in 1 mL of deionized water and weighed again for 7 consecutive days (m2). Afterwards, the samples were dried in a desiccator containing silica gel and daily weighed until a new constant mass (m3) was obtained. The values of water sorption (S) and solubility (SL), expressed in µg/mm^3^, were calculated through the formulas: S = (m2 − m3)/V and SL = (m1 − m3)/V.

### 2.9. Antibacterial Activity Analysis

The antibacterial activity analysis was performed based on a previous study methodology [16]. *Streptococcus mutans* (strain ATCC UA-159) was cultivated in TSB (tryptic soy broth) culture medium for 24 h at 37 °C in a microaerophilic environment. A total of 70 disc-shaped samples (5 mm ø × 1 mm height) were confectioned (n = 10), as described in the Water Sorption and Solubility Analyses subsection. The samples were incubated in a 96-well culture plate with 20 µL of TSB with *S. mutans* at OD_600_ 0.1476 (≅1 × 10^5^ cfu mL^−1^) for 1 h at room temperature. Afterwards, 180 µL of pure culture medium was added to the wells, and the plate was placed in an incubator for 24 h under the same conditions as those established for bacterial growth. After this period, a pipet tip was used to mix, subtract an aliquot of 100 µL from each well, and spread it on TSB-agar plates in order to verify the growth of *S. mutans*. The TSB-agar plates were also incubated for 24 h under the same conditions mentioned above. After this step, the number of colony-forming units (CFU) was quantified with the aid of a hand tally counter and converted into a percentage. Herein, it should be highlighted that samples immersed in growth culture were mixed in order to apply both planktonic and biofilm-originated bacteria to the agar plates.

### 2.10. Cytotoxicity Analysis

Cytotoxicity was evaluated through a dentin barrier test according to the principles of DIN EN ISO guidelines for in vitro cell culture model systems (I.S. EN ISO 10993 and I.S. EN ISO 7405) [29,30]. A cell line of L929 immortalized mouse fibroblast established from the normal subcutaneous areolar and adipose tissue of a male C3H/An mouse (Leibniz Institute DSMZ-German Collection of Microorganisms and Cell Cultures GmbH, Braunschweig, Germany) was cultivated with RPMI 1640 culture medium with L-glutamine supplemented with 10% fetal bovine serum (FBS) and 1% penicillin-streptomycin solution [31,32].

For sample preparation, six human third molars of approximate size were selected with approval by the local ethics committee. All teeth were freshly extracted and healthy. The teeth’s surfaces were cleaned through root scaling, followed by pumice prophylaxis application. After cleaning, the teeth were stored for no longer than 3 months in distilled water at 4 °C. For sample preparation, the tooth crowns were transversely sectioned, in their cervical third and above the occlusal limits of the pulp chambers, with a double-sided diamond disc coupled to a handpiece multi-tool (Dremel 4000, Dremel Europe, Breda, The Netherlands), under constant water cooling until 6 dentin slices were obtained. The slices were then sectioned again in 4 equal parts, generating 24 fragments, which were polished (Planopol, Struers, Ballerup, Denmark) with 800-grit silicon carbide abrasive papers under water cooling and constant speed until they reached 1 mm in height. The final size of dentin blocks was standardized at 4 × 3 × 1 mm (width × length × thickness). The block-shaped samples were allocated to the groups, including an extra one for the negative control (n = 3), so that no sample belonging to a tooth was part of the same experimental group. Customized devices made of light-bodied polyvinyl siloxane material (Honigum-Mono, DMG Chemisch-Pharmazeutische Fabrik GmbH, Hamburg, Germany) were confectioned using the samples as molds, based on a previous study [33]. The devices of 1 cm^2^ presenting an orifice of the exact size of each sample were confectioned to hold the sample floating on the culture medium and to act as a spacer between the pulpal and dentin surfaces.

Samples and devices were first disinfected with 70% isopropanol. Afterwards, the dentin was conditioned with 35% phosphoric acid (Ultra-Echt, Ultradent Products, South Jordan, UT, USA) for 15 s and rinsed for the same time with demineralized water.

Two milliliters of L929 fibroblast cell suspension (10^5^ cells/mL) of 93.2% cellular viability, stemming from the eighth passage of cell culture, was seeded in a 24-well plate. The samples held by the devices were placed in each well, floating on the culture medium with cells. A layer of adhesive, according to each experimental group, was applied to the slightly moist dentin surface, with the help of a disposable microbrush in a light scrubbing motion. The adhesive was air dried for 5 s to allow the evaporation of solvents and was then light cured (Bluephase Style, Ivoclar Vivadent, Schaan, Liechtenstein). On the negative control, instead of any adhesive, fresh culture medium was applied to the dentin surfaces. The plate was incubated at 37 °C and 5% CO_2_ for 24 h.

The samples and devices were carefully removed from the culture medium, and the cellular viability was evaluated via a colorimetric WST-1 assay. Therefore, the solutions were removed from the wells, replaced with WST-solution—96.66% culture medium + 3,33% of WST-1 proliferation reagent (2-(4-Iodophenyl)-3-(4-nitrophenyl)-5-(2,4-disulfophenyl)-2H-tetrazolium sodium salt)—and incubated for 2 h. WST-1 (red) is reduced to formazan (orange) by a reductase system present in the mitochondrial respiratory chain and is only active in metabolically intact cells. The WST-1 solutions were transferred to a new 96-well plate. Three extra wells containing the blank solution (WST-solution) were included in the test for methodology validation. The absorbance of formazan, thus cell viability, was quantified at 450 nm in a microtiter plate spectrophotometer reader (Infinite M Nano, Tecan Austria GmbH, Grödig, Austria) using Magellan Data Analysis software (https://lifesciences.tecan.com/software-magellan?p=tab--2, accessed on 23 May 2024, Tecan Trading AG, Männedorf, Switzerland). 

For data analysis, the mean blank value was subtracted from the gross values. The relative biocompatibility was expressed as the mean (%) of the negative control. The adhesives were considered to have cytotoxic potential in cases of less than 70% relative biocompatibility.

### 2.11. Statistical Analyses

Particle modification/dispersion and fracture patterns were descriptively analyzed. Remaining data were statistically analyzed (Software SPSS 21.0, SPSS Inc., Chicago, IL, USA) for normality and equity of variances using Shapiro–Wilk and Levene tests (*p* > 0.05). One-way ANOVA was performed with Tukey’s post hoc test (α = 0.05) for degree of conversion, flexural strength, modulus of elasticity, water sorption and solubility, and cytotoxicity data. Bond strength and antibacterial activity data were analyzed through a one-way ANOVA with a Fisher’s least significant difference (LSD) post hoc test (α = 0.05).

## 3. Results

### 3.1. Particles Modification/Dispersion

Figure 1 shows representative SEM micrographs of G and Gd particles, as well as microscopic images of their dispersion in the adhesives. In comparison to Gd, particles of G presented more aggregates, as pointed out by the yellow arrows. The interparticle spacing of G is therefore increased, creating gaps (red lines). The result is that G agglomerates on the adhesive margins. Presenting fewer aggregates, the spatial conformity of Gd looks more rounded and homogeneous, dispersing more effectively in the adhesive as well as in water and ethanol (Figure 2).

### 3.2. Degree of Conversion

The results of the degree of conversion are presented in Figure 3. The graph shows that, in general, the control and 0.25% Gd groups presented the highest values of degree of conversion, statistically different from the 0.5 and 0.75% G and Gd groups (*p* < 0.015), while the 0.75% Gd group obtained the lowest one. The 0.25% G group did not differ from the highest value groups or from 0.5% G (*p* > 0.05). The 0.5% groups did not differ from themselves or from 0.75% G (*p* > 0.05). The 0.5% Gd group also did not differ from 0.75% Gd (*p* > 0.05). The 0.75% groups did not differ from themselves (*p* > 0.05).

### 3.3. Flexural Strength and Modulus of Elasticity

The results of flexural strength and modulus of elasticity are shown in Table 1. The control group obtained higher flexural strength than the other groups (*p* < 0.001). The modulus of elasticity remained similar between the groups (*p* > 0.05).

### 3.4. Shear Bond Strength

The graph in Figure 4 shows that the control group presented a higher bond strength than the G groups (*p* < 0.014). The Gd groups did not differ, neither from the control nor from the G groups (*p* > 0.05).

### 3.5. Fracture Pattern

The fracture patterns are shown in Figure 5. None of the groups presented fractures of cohesive in dentin or cohesive in composite types. All groups obtained predominantly adhesive fracture between adhesive–dentin, except for 0.75% Gd, which obtained adhesive fracture between adhesive–dentin, cohesive in adhesive and mixed adhesive 1 fractures in the same proportion. Cohesive in adhesive fractures were presented only by the 0.5% and 0.75% groups. Mixed adhesive 1, which also involves cohesive failure in adhesive, was obtained by the same groups and 0.25% Gd. In general, the cohesive in adhesive fracture rate increased with the increase in G and Gd concentrations. Only G groups obtained a mixed adhesive 2 fracture pattern.

### 3.6. Water Sorption and Solubility

The results of water sorption and solubility are shown in Figure 6. The highest water sorption was obtained by the control group, with a statistical difference compared to the other groups (*p* < 0.001), except for 0.25% G (*p* > 0.05). The 0.75% G group obtained the lowest value. The 0.25% G group did not differ from the control or the 0.25% Gd (*p* > 0.05). The 0.25% Gd group, besides not differing from 0.25% G, also did not differ from 0.5% groups (*p* > 0.05). The 0.5% groups did not differ from themselves, and 0.5% G also did not differ from the 0.75% ones (*p* > 0.05). The 0.5% Gd group did not differ from 0.75% Gd (*p* > 0.05). The 0.75% groups did not differ from themselves (*p* > 0.05).

The highest solubility was obtained by 0.25% G, and the lowest was obtained by 0.75% Gd. The 0.5% G group did not differ from any other group (*p* > 0.05). The control group also did not differ from any other group (*p* > 0.05), except 0.75% Gd (*p* = 001). The 0.75% G, 0.25% Gd, and 0.5% Gd did not differ from any other group (*p* > 0.05), except 0.25% G (*p* < 0.032).

### 3.7. Antibacterial Activity

The graph in Figure 7 shows that the control and 0.25% G groups presented the highest percentage of Colony-forming units (*p* < 0.001), while the 0.75% groups presented the lowest ones. The 0.5% G and 0.25% Gd groups did not differ from themselves (*p* > 0.05). The 0.5% Gd group did not differ from other Gd groups or from 0.75% G (*p* > 0.05). The 0.75% groups did not differ from themselves (*p* > 0.05).

### 3.8. Cytotoxicity

Figure 8 shows the cytotoxicity results. Negative control presented the highest value of relative biocompatibility, while control, 0.25%, and 0.5% groups presented the lowest ones and were considered to have cytotoxic potential. The 0.75% groups did not differ from other groups (*p* > 0.05).

## 4. Discussion

Dispersing the particles in a polymer-based matrix, such as from dental adhesives, is naturally complicated due to unfavorable interactions between the matrix and the nanoparticles [34,35]. The results of this study showed that graphene modified by L-DOPA was better dispersed than non-modified graphene. Graphene is a two-dimensional monolayer of sp^2^ carbon atoms tightly bonded to each other in a hexagonal lattice [10,12,13]. In the case of its particles, specifically, their stabilized dispersion is challenging due to synthesis-related entanglement, lower surface energy, and a high intersheet van der Waals force [17,18,19,36]. 

Based on the Derjaguin–Landau–Verwey–Overbeek (DLVO) theory, particles should be stable in fluids if repulsive forces are dominant over attractive forces between particles [37]. In general, the strategies addressed by the literature to disperse and stabilize nanoparticles include electrostatic and steric stabilization [34,38]. In the first one, the attractive forces between particles are counterbalanced by repulsive forces. In the later one, polymer chains cover the nanoparticle surface, functionalizing it, so when the particles aggregate, the polymer segments also approach, reducing the system’s extent of molecular freedom and increasing the so-called Gibbs free energy. As a result, the agglomeration of particles is inhibited [34]. Graphene particles may be dispersed in both ways [39]. The structure of graphene provides a large surface area to be functionalized, leading to steric-based stabilization [12]. This functionalization may occur, i.e., through non-covalent π–π interactions with hydrophobic molecules or, covalently, through the sharing of sp2 orbitals [12,13,40]. The functionalized graphene can then be further stabilized through electrostatic forces, generating electrosteric stabilization [38,39].

To the best of our knowledge, there is no report in the literature about the functionalization of graphene with L-DOPA. However, we may infer that, thanks to its amphiphilic feature, L-DOPA’s aromatic ring establishes π-interactions with the basal plane of graphene, while its o-hydroxyl groups engage with the bulk aqueous medium [20,40]. As a result, dispersibility is enhanced. In the adhesive system, specifically, L-DOPA still interacts hydrophobically with alkyl groups of methacrylates through its aromatic ring, and depending on the ζ potential of the modified graphene, the particles should be further stabilized electrostatically against aggregation [20,39]. Evidence of the stabilized dispersion of graphene by L-DOPA is shown by Figure 1 and Figure 2. In the SEM image, little indication of defects is seen among the modified particles. Conversely, non-modified graphene presents agglomerations of particles, which lead to their poor dispersion in aqueous medium.

The results of particle dispersion may justify further results of this study. The control and 0.25% Gd groups presented the highest values of degree of conversion. Although graphene presents 2.3% of linear optical absorption, making it nearly virtually transparent to light [10,17,40], this feature is only identified when graphene is obtained in a single layer, such as through the scotch tape method [9]. In fact, the accurate obtainment of graphene single layers from graphite cleaving remains a major challenge [10]. Accordingly, graphene is mostly obtained in multiple layers, turning it dark and reducing its ability to transmit light [12]. Analyzing our findings, one may infer that the higher the amount of graphene, the less light is transmitted by the whole adhesive layer and the higher the light scattering. This is reasonable, once the way light is transmitted through the adhesive layer during light curing is influenced by several factors: filler content and distribution, type and size of fillers, light source, and difference of refractive index between fillers and monomers [41].

A graphene-based adhesive may have its degree of conversion affected first—because dark pigmented particles tend to absorb more light, decreasing the depth of cure [42]; second—because the higher the particle size, the higher the scattering [41]—although the graphene used in this study was composed of sub-micronparticles, their natural agglomeration makes them behave as units of larger size; and last—due to natural chemistry interactions between the graphene and monomers. During polymerization, monomer conversion is never complete [43]. A percentage of monomer remains unreacted due to diffusion limitations of the organic components, which lead to the gel effect [44]. One may suppose that graphene particularities can hinder such diffusion. This explains why groups with a higher amount of particles presented the lowest percentage of degree of conversion. Yet, it is important to bear in mind that degree of conversion is only one of various predictors of the quality of polymerization [45] and that the rate of degree of conversion of dental composite resins may naturally present great variation (from 35% to 77%) [44]. Thus, the applicability of the adhesive with 0.75% particles would not be impracticable based on these results alone. 

Nonetheless, it is fair to acknowledge that the stabilized particle dispersion in the 0.25% Gd group and possible lesser light scattering led to a similar degree of conversion compared to the control group. It is plausible that the similar bond strength of Gd groups compared to the control may also be in part explained by the better degree of conversion in the systems where there is a better dispersion of particles. In addition, we may infer that the action of L-DOPA as a chelating agent, interacting strongly with organic or inorganic substrates, could perhaps be considered as a further factor that justifies the similar Gd and Control bond strength results [21,22,23].

Still related to particle dispersion, the antibacterial activity test showed that the 0.25% Gd group did not differ from 0.5% G, and the 0.5% Gd group did not differ from 0.75% G. Concomitantly, 0.25% G presented similar antibacterial performance to the control. Although it was already reported that aggregated graphene in suspensions may trap and isolate bacterial cells from their microenvironment [15], our findings suggest that stabilized particle dispersion and consequently restacking prevention may be directly proportional to the enhancement of graphene’s antibacterial activity. Moreover, once it is amphiphilic, L-DOPA might have established hydrophobic interactions with lipid layers of bacterial membranes, extracting phospholipids and dissolving them further; and possibly, during oxidation, L-DOPA may have generated reactive oxygen species (ROS) that, together with graphene’s ROS, act as biocides [46].

The fact that 0.75% groups presented the lowest rate of CFU provides evidence that the antibacterial efficiency of graphene is concentration-dependent, which is consistent with previous reports [14]. As well stated by Al-Jumaili et al., the diverse intrinsic properties of graphene make it difficult to predict its exact mechanism of antimicrobial action, mainly when other compounds are present [15]. Nonetheless, it is known that graphene is able first—to penetrate and cut the cell membrane of microorganisms; second—to wrap cells, causing mechanical stress; third—to extract phospholipids as a result of chemical interaction between the hydrophobicity of graphene and bacterial lipid layers; fourth—to produce ROS, leading the bacteria cells to the oxidative stress state and harming their components; and last—to attract electrons from the microbial membrane, compromising its integrity [13,14,47]. With *Streptococcus mutans*, specifically, graphene is also related to the antifouling effect [16]. 

In our study, by adding L-DOPA to the adhesive’s system, the antifouling effect could also possibly be improved due to an increase in hydrophilicity [34,48]. Nevertheless, Figure 6 indicates that L-DOPA had a slight impact on the adhesive’s hydrophilicity. The graph of water sorption shows that 0.25% Gd and 0.5% Gd are likely to present more constant values, but in fact, except for 0.25% G, which was similar to the control group, all groups presented overall similar behavior, with a tendency for water sorption to decrease with G/Gd concentration increase. Likewise, in solubility, groups behaved in the same way, and 0.25% G is once more noteworthy, as it presented the highest value, opposing 0.75% Gd, which showed the lowest one. One may infer that, in the case of 0.25% G, the lowest concentration of graphene and the agglomeration of non-modified particles led to a hydrophobicity decrease. In general, the extent and rate of water sorption, solubility, and diffusion increased with the material’s hydrophilicity [49]. The results suggest that the addition of L-DOPA did not increase enough the adhesive’s hydrophilicity but rather improved the availability of graphene through better dispersion of particles, improving also the hydrophobicity characteristic promoted by graphene.

The antibacterial activity of G and Gd is remarkable, but there is concern about cytotoxicity. When a compound is active against microbes, it is likely to be active in other body cells [34]. Analyzing the graph of Figure 8, although all adhesives presented statistical similarity, it is worthy of attention that the only groups that were also statistically similar to the negative control (fresh culture medium) were the 0.75% ones. Considering the dentin barrier test simulates an in vivo oral environment and factors that affect diffusion through dentinal tubules are taken into account [50], our findings suggest that either the higher amount of graphene impairs proper adhesive diffusion through the dentinal tubules or graphene benefits the growth of fibroblast-like cells. The first hypothesis could be reasonable if we consider the reports from Bregnocchi and colleagues, who showed that adhesives with a higher content of graphene are not well exposed when substrates with small porosities are tested [16]. On the other hand, if the adhesives were not diffusing properly through the dentinal tubules, we would also expect a lower bond strength of 0.75% groups, which did not occur. Thus, we are prone to accept the second hypothesis, especially considering various previous studies reported similar phenomena to ours, including other cell types [51,52,53,54,55,56]. Lasocka et al., i.e., revealed that graphene is not only non-toxic to L929 fibroblast cells but also increases their proliferation and mitochondrial activity [51]. With no signs of cytotoxicity, graphene was additionally found to increase human mesenchymal stem cell proliferation and lead to differentiation into dental tissues, such as osteoblasts [57]. Indeed, accumulating studies indicate that the cytotoxicity of graphene depends on the concentration, shape, size, conformation, dispersibility, and functionalization of particles, making it difficult to state its toxicity level [12,47]. In light of the above, it appears, however, that graphene’s propensity for biocompatibility is worthy of attention.

The high modulus of elasticity of graphene has been extensively reported in the literature [10,11,12,13]. In this study, the modulus of elasticity did not differ between the adhesives. It is possible that the amount of G and Gd was not high enough to increase such values. Nevertheless, the results of the fracture pattern still indicate that the higher the concentration of G and Gd, the higher the mechanical strength due to the large regular arrangement of carbon atoms joined by covalent bonds [10,13,58]. The 0.5% and 0.75% groups presented cohesive in adhesive fracture, either alone or associated with other fracture types (mixed). In the last case, mixed fractures involving cohesive in adhesive fracture were also shown by 0.25% Gd, pointing out once more that proper dispersion of graphene improves its features. Yet, although graphene’s mechanical strength is high and, incorporated in adhesives, may lead to cohesive fractures, in this study, its adhesives presented lower flexural strength than the control, as shown by Table 1. The values are correlated to the high flexibility feature presented by graphene. As well reported by Lee and colleagues, graphene presents nonlinear elastic behavior, and may undergo large elastic deformations without requiring much energy. Namely, graphene has zero bending stiffness [59]. Such characteristics, added to their high strength, certainly pose an opportunity for graphene to be applied in dental materials, where absorption and release of occlusal loads should be considered. 

This study still has limitations that cannot be neglected. Graphene’s dark color remains one of the major drawbacks of its addition to the adhesive system. Considering esthetics are increasingly valued in restorative dentistry, this feature is a concern. In addition, this is an in vitro study. Caution must be exercised when applying novel in-vitro results to clinical practice.

## 5. Conclusions

By modifying graphene with L-DOPA, graphene properties, and therefore the properties of adhesives incorporated by it, were in general enhanced. A total of 0.75 wt% of Gd incorporated in the dental adhesive kept the bond strength and modulus of elasticity similar to the ones of the control group, while improving antibacterial activity and decreasieng cytotoxicity, flexural strength, water sorption, and solubility. The results presented here point out that DOPA-modified graphene represents a promising compound to be added in dental adhesive.

## Figures and Tables

**Figure 1 polymers-16-02081-f001:**
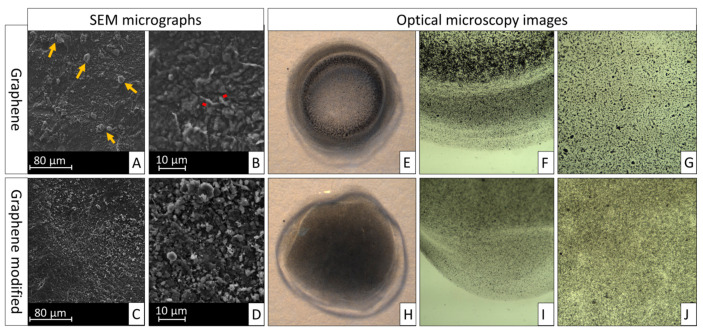
Representative scanning electron micrographs of graphene and graphene modified by L-DOPA surfaces and optical microscopy images of their particle dispersion in the adhesives. Yellow arrows show particles aggregates and red lines show interparticle spacing. (**A**) Graphene (1000×); (**B**) graphene (5000×); (**C**) graphene modified by L-DOPA (1000×); (**D**) graphene modified by L-DOPA (5000×); (**E**) 0.5% G adhesive visible to the naked eye; (**F**) peripheral image of 0.5% G adhesive; (**G**) central image of 0.5% G adhesive; (**H**) 0.5% Gd adhesive visible to the naked eye; (**I**) peripheral image of 0.5% Gd adhesive; (**J**) central image of 0.5% Gd adhesive.

**Figure 2 polymers-16-02081-f002:**
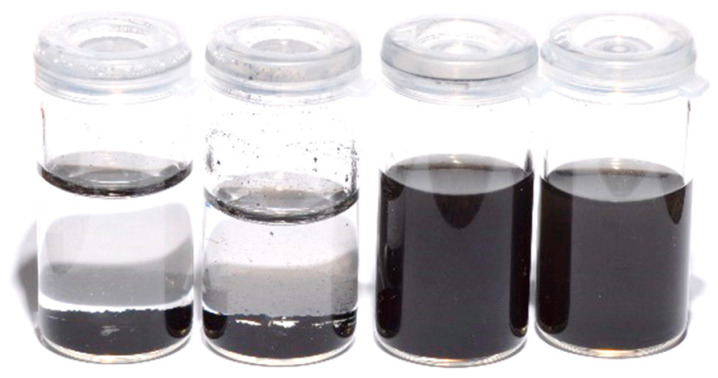
From left to right—2 mg/mL of G dispersed in deionized water, 2 mg/mL of G dispersed in ethanol, 2 mg/mL of Gd dispersed in deionized water, 2 mg/mL of Gd dispersed in ethanol.

**Figure 3 polymers-16-02081-f003:**
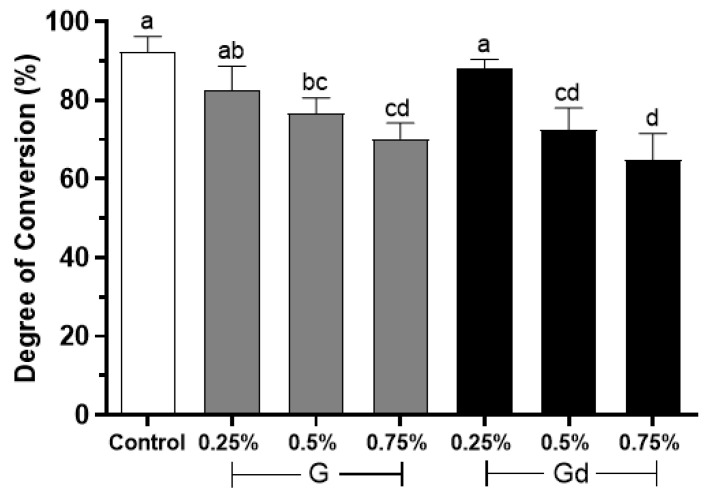
Degree of conversion (%) of experimental adhesives. Distinct letters indicate statistical difference (*p* ≤ 0.05), with a being statistically higher than b, b being statistically higher than c, and c being statistically higher than d. Two letters placed together (e.g., ab) indicate that the value obtained by the group does not differ statistically from values identified by one (a) or the other letter (b). Error bar indicates standard deviation.

**Figure 4 polymers-16-02081-f004:**
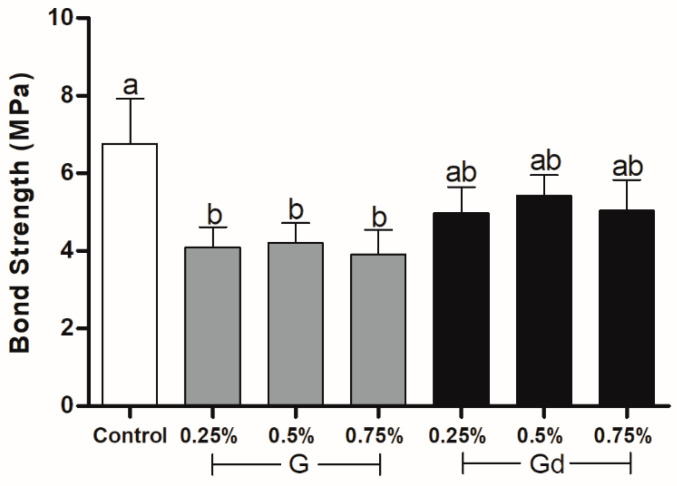
Bond strength (MPa) of experimental adhesives. Distinct letters indicate statistical difference (*p* ≤ 0.05), with a being statistically higher than b. Two letters placed together (e.g., ab) indicate that the value obtained by the group does not differ statistically from values identified by one (a) or the other letter (b). Error bar indicates standard deviation.

**Figure 5 polymers-16-02081-f005:**
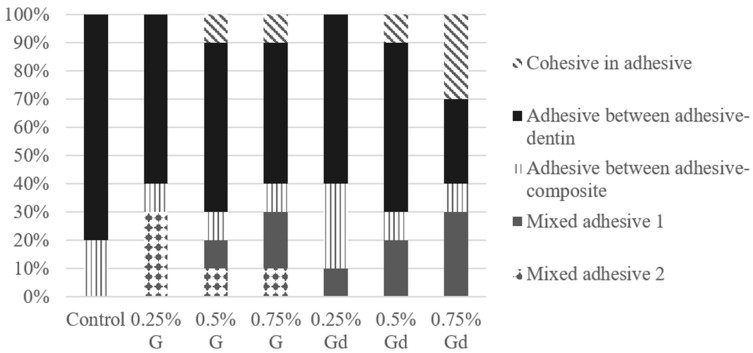
Facture pattern (%) of experimental adhesives.

**Figure 6 polymers-16-02081-f006:**
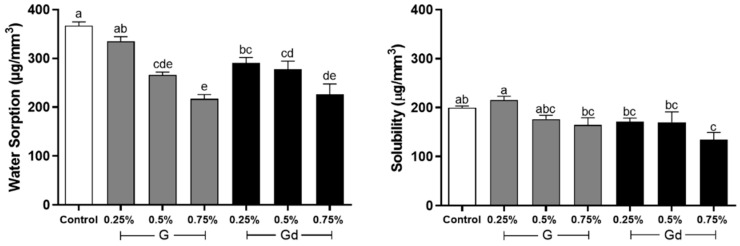
Water sorption and solubility (µg/mm^3^) of experimental adhesives. Distinct letters indicate statistical difference (*p* ≤ 0.05), with a being statistically higher than b, b being statistically higher than c, c being statistically higher than d, and d being statistically higher than e. Two or three letters placed together (e.g., ab) indicate that the value obtained by the group does not differ statistically from values identified by one (a) or the other letter (b). Error bar indicates standard deviation.

**Figure 7 polymers-16-02081-f007:**
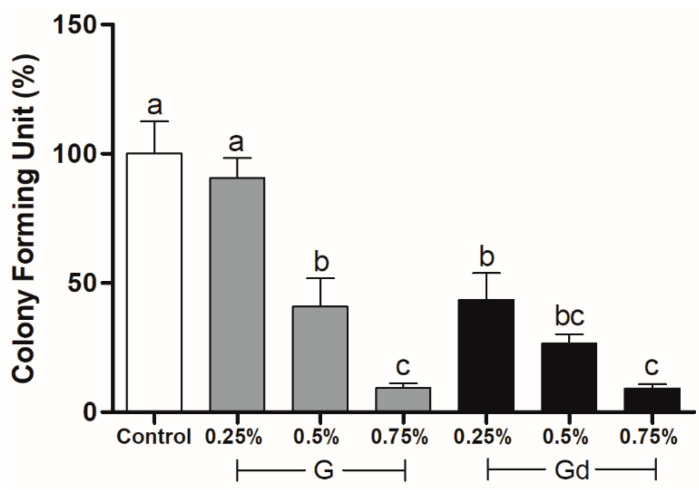
*Streptococcus mutans* colony-forming unit (%) on experimental adhesives. Distinct letters indicate statistical difference (*p* ≤ 0.05), with a being statistically higher than b, and b being statistically higher than c. Two letters placed together (e.g., bc) indicate that the value obtained by the group does not differ statistically from values identified by one (b) or the other letter (c). Error bar indicates standard deviation.

**Figure 8 polymers-16-02081-f008:**
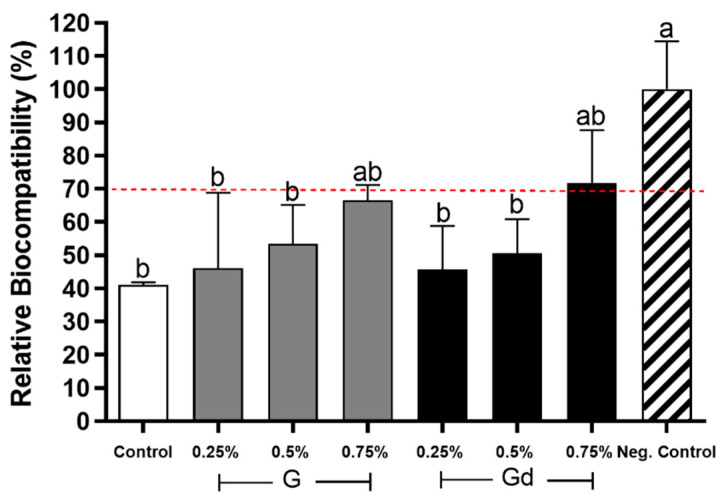
Relative biocompatibility (%) of experimental adhesives. Negative control: fresh culture medium. Distinct letters indicate statistical difference (*p* ≤ 0.05), with a being statistically higher than b. Two letters placed together (e.g., ab) indicate that the value obtained by the group does not differ statistically from values identified by one (a) or the other letter (b). Error bar indicates standard deviation.

**Table 1 polymers-16-02081-t001:** Mean (standard deviation) flexural strength and modulus of elasticity (MPa) of experimental adhesives. Mean values followed by distinct letters differ statistically at 5%, according to one-way ANOVA and Tukey post hoc test.

Group	Flexural Strength	Modulus of Elasticity
Control	17.08 (4.9) ^a^	237.34 (59.1) ^a^
0.25% G	5.48 (1.8) ^b^	197.59 (45.2) ^a^
0.50% G	5.28 (1.1) ^b^	210.08 (51.6) ^a^
0.75% G	3.60 (0.8) ^b^	170.92 (69.9) ^a^
0.25% Gd	5.41 (1.8) ^b^	162.94 (33.1) ^a^
0.50% Gd	4.06 (0.3) ^b^	231.40 (78.4) ^a^
0.75% Gd	4.68 (2.4) ^b^	264.28 (100.1) ^a^

Mean values followed by distinct letters differ statistically at 5%, according to one-way ANOVA and Tukey post hoc test, with a being statistically higher than b.

## Data Availability

Source data are available on request from the corresponding author.

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
