# Peer review of "Properties of a Dental Adhesive Containing Graphene and DOPA-Modified Graphene"

_polymers, 2024, doi:10.3390/polym16142081_

Round 1

Reviewer 1 Report

Comments and Suggestions for Authors

This is an interesting and comprehensive study on the effects of doping an experimental dental adhesive with graphene and DOPA-modified graphene. The multiple benefits of the doped adhesives were clearly shown but I still have several issues for the authors to consider:

 1. “Conversion degree” – please change the term to “degree of conversion” – this term is much more common

2. Introduction: “The findings of this paper may repre-80 sent a significant step on the graphene-based materials study for dental and medicine applications. As part of our highlights, we found out first – that L-dopa is a catechol that can 82 disperse graphene particles; second – that properties of graphene are enhanced with its modification with L-dopa; and last – that graphene-dopa-based dental adhesive promote antibacterial activity, improve biocompatibility, mechanical strength and keeps the material flexible to absorb tensions.” - Please remove the results of the present study from the introduction section!!!

3. Materials and methods, section 2.4. Conversion degree analysis: “From this test onwards, the entire methodology was carried out according to ISO 142 4049 [26]” - The ISO 4049 protocol does not include analysis of the degree of conversion by FTIR (nor with any other spectroscopic technique). Please revise this.

4. Materials and methods: “The peaks located at 1638 cm-1 (aliphatic C=C bond) 157 and 1608 cm-1 (aromatic component group) bands were used as internal standard [28].” - Only the latter peak is used in the FTIR analysis as an internal standard, while the sentence suggests that both peaks were internal standards. Please rephrase.

5. Materials and methods: “Samples were tested through 3-point flexural test using a universal mechanical test machine” - This test is designed for resin composites, which are far stiffer than adhesives. Please discuss whether the mechanics of the test are relevant to highly elastic materials such as your experimental adhesives. The load-displacement curves must have had a non-linear part and a lot of yielding before fracture. Should the flexural strength be evaluated for such specimens, or would it perhaps be more appropriate to evaluate the yield strength? Please also define from which part of the load-displacement curve the modulus of elasticity was determined, as this also assumes a linear part of the curve, which was certainly not completely linear.

6. “Microshear bond strength testing” – specimens with a diameter of 2.32 cannot be considered micro-shear. It is macro-shear. Please see 10.1016/j.dental.2009.11.148.

7. Results: Figure 4 - The mean value for the control material seems quite low. Why did you choose to formulate your own adhesive instead of doping a commercially available adhesive with graphene? This could lead to better values for bond strength and thus bring the results a little closer to clinical relevance.

8. Results: Figure 5 - How did you define the failure mode "cohesive in adhesive"? It is usually not evaluated as such because the adhesive layer is very thin and the specimens that fractured cohesively within the adhesive failed are deemed as "adhesive failures".

9. Discussion: “The action of L-DOPA as a chelating agent, 540 interacting strongly with organic or inorganic substrates, should then be considered as a 541 further factor that justifies the bond strength results [21-23].” - This is only speculation and is not supported by the data of the present study as you don't have a statistically significant improvement between the doped and non-doped graphene.

10. Discussion: “Graphene’s dark color re-622 mains one of the major drawbacks of its addition in the adhesive system.” - I could not agree more. Dark colored adhesives would never be accepted in clinical practice. Please also amend the Conclusions section where it is stated that graphene is likely to be added to restorative materials. Not only is this unlikely, it is also not a conclusion of the present study (there is no data to support the claim about adhesives being accepted in clinical practice).

Author Response

  1. “Conversion degree” – please change the term to “degree of conversion” – this term is much more common

Thank you very much. The term was modified in the whole text, accordingly. The abstract was slightly changed, in order to keep the maximum of 200 words allowed by the journal. In addition, Figure 3 was replaced by a new one with the corrected heading (Page 8, Line 350).

  1. Introduction: “The findings of this paper may represent a significant step on the graphene-based materials study for dental and medicine applications. As part of our highlights, we found out first – that L-dopa is a catechol that can disperse graphene particles; second – that properties of graphene are enhanced with its modification with L-dopa; and last – that graphene-dopa-based dental adhesive promote antibacterial activity, improve biocompatibility, mechanical strength and keeps the material flexible to absorb tensions.” - Please remove the results of the present study from the introduction section!!!

Thank you for the suggestion. The main conclusions of our study were included in the Introduction section, as it was part of the instructions for the manuscript preparation ("Finally, briefly mention the main aim of the work and highlight the main conclusions” - 4th line of Research Manuscript Sections heading, Introduction subheading). Nevertheless, they were removed from the paragraph, as requested.

  1. Materials and methods, section 2.4. Conversion degree analysis: “From this test onwards, the entire methodology was carried out according to ISO 142 4049 [26]” - The ISO 4049 protocol does not include analysis of the degree of conversion by FTIR (nor with any other spectroscopic technique). Please revise this.

Thank you for pointing this out. We agree. We replaced the reference in another sentence, in the beginning of “Flexural strength and modulus of elasticity analyses” subsection (Page 4, Lines 156-157). The text is now accurate. As the references have been reordered, the list at the end of the manuscript has also been slightly modified (Page 16, Lines 695-701).

  1. Materials and methods: “The peaks located at 1638 cm-1 (aliphatic C=C bond) 157 and 1608 cm-1 (aromatic component group) bands were used as internal standard [28].” - Only the latter peak is used in the FTIR analysis as an internal standard, while the sentence suggests that both peaks were internal standards. Please rephrase.

Thank you for the observation. The sentence was rephrased (Page 3, Lines 149-150).

  1. Materials and methods: “Samples were tested through 3-point flexural test using a universal mechanical test machine” - This test is designed for resin composites, which are far stiffer than adhesives. Please discuss whether the mechanics of the test are relevant to highly elastic materials such as your experimental adhesives. The load-displacement curves must have had a non-linear part and a lot of yielding before fracture. Should the flexural strength be evaluated for such specimens, or would it perhaps be more appropriate to evaluate the yield strength? Please also define from which part of the load-displacement curve the modulus of elasticity was determined, as this also assumes a linear part of the curve, which was certainly not completely linear.

Thank you for bringing out this concern. At the time of our study, only one paper addressing graphene incorporated into a dental adhesive had been published. Several properties about this topic could still be investigated and explored. As graphene typically presents high mechanical strength, we were not sure how the adhesives incorporated by it would perform mechanically, i.e. if the particles concentrations we added would make the adhesives stiffer. Therefore, we considered flexural strength could be an appropriate test. Also, the test provided us the modulus of elasticity of the samples. Both properties can give us an estimate of how the material would perform clinically. From our results, for example, we can imagine that in a high c-factor cavity, where the polymerization shrinkage stress is high, the low flexural strength of the adhesives incorporated with graphene and dopa-modified graphene would facilitate the absorption and dissipation of stress. Indeed, evaluating the yield strength could also be relevant to our study and would provide additional important data about the adhesives’ elastic behavior. For further investigation, we may certainly consider evaluating the yield strength. The modulus of elasticity was determined from the slope of the linear portion of the load-deflection curve. We reworded this part of the test description and added the information in the Flexural strength and modulus of elasticity analyses subsection (Page 4, Lines 166-172).

  1. “Microshear bond strength testing” – specimens with a diameter of 2.32 cannot be considered micro-shear. It is macro-shear. Please see 10.1016/j.dental.2009.11.148.

Thank you for pointing this out. We have modified the term to shear in the whole text.

  1. Results: Figure 4 - The mean value for the control material seems quite low. Why did you choose to formulate your own adhesive instead of doping a commercially available adhesive with graphene? This could lead to better values for bond strength and thus bring the results a little closer to clinical relevance

We appreciate your concern. We believe that handling a commercial adhesive could generate methodological biases, given that it is a product formulated by the manufacturer and that full knowledge of the chemical composition of the compounds included, as well as their concentration, may be limited. Additionally, the initial idea of this study would be to patent our formulation, depending on the results obtained. Therefore, we did not consider it appropriate to simply add the graphene and modified graphene to a formulation that was already commercially available.

  1. Results: Figure 5 - How did you define the failure mode "cohesive in adhesive"? It is usually not evaluated as such because the adhesive layer is very thin and the specimens that fractured cohesively within the adhesive failed are deemed as "adhesive failures".

Thank you for pointing this out. The fracture patterns were evaluated both through visual analysis and by optical microscopy. As the graphene-added adhesives were black, we could observe remaining adhesive either on the dentin surface, on the composite surface, or on both. For cohesive in adhesive failure mode, we could see an adhesive layer both on the dentin and on the composite pillar surface, as you can check in the picture below.

Figure 1. Sample submitted to shear bond strength test showing cohesive in adhesive fracture pattern.

  1. Discussion: “The action of L-DOPA as a chelating agent, 540 interacting strongly with organic or inorganic substrates, should then be considered as a 541 further factor that justifies the bond strength results [21-23].” - This is only speculation and is not supported by the data of the present study as you don't have a statistically significant improvement between the doped and non-doped graphene.

Through this sentence, followed by the previous one ("It is plausible that the similar bond strength of Gd groups compared to Control...") our intention was to provide an explanation of the possible reasons why the Gd group showed similar bond strength to the Control group. Nevertheless, we agree that the statement is only speculation. We have reworded the sentence to make this clear (Page 12, Lines 511-514). Thank you very much.

  1. Discussion: “Graphene’s dark color re-622 mains one of the major drawbacks of its addition in the adhesive system.” - I could not agree more. Dark colored adhesives would never be accepted in clinical practice. Please also amend the Conclusions section where it is stated that graphene is likely to be added to restorative materials. Not only is this unlikely, it is also not a conclusion of the present study (there is no data to support the claim about adhesives being accepted in clinical practice).

Thank you for the suggestion. We removed the sentence from the Conclusions section.

Reviewer 2 Report

Comments and Suggestions for Authors

Dear authors:

Following my review of your article “Properties of a dental adhesive containing graphene and 2 DOPA-modified graphene”, I would like to share some comments.

This is a well-written article that addresses a pertinent topic, particularly to the field of restorative dentistry and adhesive dentistry. The writing is clear, concise, and accurate, making it easy to understand the content and the methodological design is very complete, which strengthens the reliability of the results. The results are interesting and relevant to the research topic.

In my opinion, it should be considered for publication.

Author Response

Following my review of your article “Properties of a dental adhesive containing graphene and 2 DOPA-modified graphene”, I would like to share some comments.

This is a well-written article that addresses a pertinent topic, particularly to the field of restorative dentistry and adhesive dentistry. The writing is clear, concise, and accurate, making it easy to understand the content and the methodological design is very complete, which strengthens the reliability of the results. The results are interesting and relevant to the research topic.

In my opinion, it should be considered for publication.

Thank you very much for taking the time to review this manuscript. The authors are truly thankful for your consideration and compliments.

Reviewer 3 Report

Comments and Suggestions for Authors

Properties of a dental adhesive containing graphene and DOPA-modified graphene

The article investigates the modification of graphene nanoparticles with L-DOPA to enhance the properties of dental adhesives. Experimental adhesives were formulated with different concentrations of modified and unmodified graphene. The study assessed particle dispersion, conversion degree, flexural strength, bond strength, water sorption, solubility, antibacterial activity, and cytotoxicity. Results showed that L-DOPA modification improved graphene dispersion, leading to enhanced adhesive properties. Adhesives with modified graphene exhibited similar bond strength to the control and reduced water sorption and solubility. Additionally, less modified graphene was needed for antibacterial effectiveness. The study concludes that L-DOPA-modified graphene enhances the mechanical, antibacterial, and biocompatibility properties of dental adhesives.

Comments:

1.     What challenges are associated with the dispersion of graphene in aqueous mediums, and how does the modification with L-DOPA address these challenges?

2.     What were the main components used in formulating the experimental dental adhesive in this study?

3.     How was the flexural strength and modulus of elasticity of the adhesives tested, and what were the results?

4.     Use the following references to deepen the introduction and ANOVA. Various FDM mechanisms used in the fabrication of continuous-fiber reinforced composites: a review. Shape memory performance assessment of FDM 3D printed PLA-TPU composites by Box-Behnken response surface methodology.

5.     What methodology was used to test the antibacterial activity of the adhesives, and what were the findings?

6.     How did the modification of graphene with L-DOPA impact the water sorption and solubility of the adhesives?

7.     What were the main findings regarding the cytotoxicity of the adhesives on fibroblasts?

8.     Describe the process and significance of using Fourier-transform infrared spectroscopy (FTIR) in analyzing the conversion degree of the adhesives.

9.     How did the bond strength of the adhesives compare between the control group and those with varying concentrations of graphene and graphene modified by L-DOPA?

10.  What were the significant conclusions drawn about the properties of graphene and its modification with L-DOPA in the context of dental adhesives?

Comments on the Quality of English Language

***

Author Response

  1. What challenges are associated with the dispersion of graphene in aqueous mediums, and how does the modification with L-DOPA address these challenges?

Thank you very much for taking the time to review this manuscript. The main challenge associated with the dispersion of graphene in aqueous mediums is particles agglomeration. It happens due to graphene hydrophobic nature, lower surface energy and high van der Waals force between its layers. The L-DOPA might work as an amphiphilic dispersant, which reduces solvent’s surface energy, promotes exfoliation, and prevents particles re-aggregations. In addition, L-DOPA may act as a chelating agent, interacting strongly with organic or inorganic substrates. Therefore, they are able to promote nanoparticle stabilization.

  1. What were the main components used in formulating the experimental dental adhesive in this study?

The components used in our formulation were 7 wt% water; 30 wt% ethanol, 20 wt% Bisphenol A glycidyl methacrylate (Bis-GMA), 5 wt% Diurethane dimethacrylate (UDMA); 15 wt% 2-Hydroxyethyl methacrylate (HEMA); 10 wt% Glycerol 1,3-dimethacrylate; 10 wt% Polyacrylic acid; 1 wt% Ethyl-4-Dimethylaminobenzoate (EDAB); 1,5 wt% Diphenyliodonium hexafluorophosphate (DPIHP); and 0,5 wt% Camphorquinone. This experimental adhesive received then the addition of 0.25 wt%, 0.5 wt% or 0.75 wt% of graphene or graphene modified by L-DOPA.

  1. How was the flexural strength and modulus of elasticity of the adhesives tested, and what were the results?

A total of 35 bar-shaped samples (n=5) were confectioned using a mould of heavy and light-bodied polyvinyl siloxane material. The adhesives were applied in the mould until complete filling and were covered by a polyester strip. Each sample was light cured on top and bottom surfaces, on both extremities and on the middle. The samples were tested through 3-point flexural test using a universal mechanical test machine at 1 mm/min speed with 50 N load. The flexural strength (FS) and modulus of elasticity (ME), expressed in MPa, were calculated through the formulas: FS = 3FL/2bh2; and ME = FL3/4bh3d, where F is the force required to cause the sample failure, in Newton, L is the distance between the supports, b is the sample width, h is the sample thickness, and d is the load-deflection, all in millimeters. Control group obtained higher flexural strength than the other groups. The modulus of elasticity remained similar between the groups.

  1. Use the following references to deepen the introduction and ANOVA. Various FDM mechanisms used in the fabrication of continuous-fiber reinforced composites: a review. Shape memory performance assessment of FDM 3D printed PLA-TPU composites by Box-Behnken response surface methodology.

Thank you for your suggestion. Both articles are very interesting, but with all due respect, we do not understand the relationship we could create between 3d printing through Fused Deposition Modeling with the topic of our article, which is dental adhesive (typically used during a clinical restorative procedure), added by graphene and graphene modified by L-DOPA. We believe that the references are unfortunately not relevant to the content of our manuscript and it would be difficult to create a connection to include them in our text.

  1. What methodology was used to test the antibacterial activity of the adhesives, and what were the findings?

We appreciate your concern. For antibacterial activity analysis, Streptococcus mutans bacteria was cultivated in Tryptic Soy Broth (TSB) culture medium for 24 hours at 37°C under microaerophilic environment. A total of 70 disc-shaped samples (5 mm ø x 1 mm height) were confectioned (n=10). The samples were incubated in a 96-well culture plate with 20 µl of TSB with S. mutans at OD600 0.1476 for 1 hour at room temperature. Afterwards, 180 µl of pure culture medium were added to the wells, and the plate was placed in an incubator for 24 hours under the same conditions as those established for bacterial growth. After this period, a pipet tip was used to mix, subtract an aliquot of 100 µl from each well and spread it on TSB-agar plates, in order to verify the growth of S. mutans. The TSB-agar plates were also incubated for 24 hours under the same conditions mentioned above. After this step, the number of Colony Forming Units (CFU) was quantified with the aid of a hand tally counter and converted into percentage. Control and 0.25% G groups presented the highest percentage of Colony Forming Units, while 0.75% groups presented the lowest ones. The 0.5% G and 0.25% Gd groups did not differ from themselves. The 0.5% Gd group did not differ from other Gd groups and from 0.75% G. The 0.75% groups did not differ from themselves.

  1. How did the modification of graphene with L-DOPA impact the water sorption and solubility of the adhesives?

Thank you for your concern. In general, the extent and rate of water sorption, solubility and water diffusion is increased with the material’s hydrophilicity. We believed that, by adding L-DOPA to the adhesive’s system, the water sorption and solubility could get higher due to increase of hydrophilicity. Nevertheless, all groups presented overall similar behavior, with a tendency of water sorption decrease with increase of particles, both graphene and L-DOPA-modified graphene, which is very interesting for the purpose of dental adhesives. The results suggest that the addition of L-DOPA did not increase enough the adhesive’s hydrophilicity, but rather improved the availability of graphene through better dispersion of particles, improving also the hydrophobicity characteristic promoted by graphene.

  1. What were the main findings regarding the cytotoxicity of the adhesives on fibroblasts?

Negative control presented the highest value of relative biocompatibility, while Control, 0.25% and 0.5% groups presented the lowest ones and were considered to have cytotoxic potential. The 0.75% groups did not differ from other groups.

  1. Describe the process and significance of using Fourier-transform infrared spectroscopy (FTIR) in analyzing the conversion degree of the adhesives.

Thank you for your question. Specifically for the analysis of degree of conversion of methacrylate-based resins, as the adhesives, the spectra are usually obtained in the 4000-650 cm-1 range, using 64 scans at 4 cm-1 resolution in transmission mode and 2.8 mm/s speed (all these parameters can be slightly varied depending on the FTIR equipment used). The height of the absorbance peak is determined after subtraction of the baseline and the normalization process, using FTIR software or other one such as Origin. The intensities of the absorbance peaks located in the 1638 cm-1 band of each spectrum correspond to the identification of aliphatic carbon chains (C = C), and in the 1608 cm-1 band, to the presence of aromatic carbon chains. Analysis of the latter band serves as internal standard for calculating the percentage of remaining carbon double bonds in adhesives after curing. Thus, the percentage of degree of conversion (DC) is calculated using the following formula: DC (%) = 100 x [1 – (R cured/R uncured)], where R represents the ratio between aliphatic band absorption at 1638 cm−1 and aromatic band absorption at 1608 cm−1, respectively. Analysis of degree of conversion of monomeric materials – such as those used in this research – using FTIR is widely used and validated in dental research, as it contributes to the characterization of experimental materials. In this study, it was important to investigate whether graphene and DOPA-modified graphene would impair the degree of conversion of adhesives, since poorly cured adhesives generally degrade quickly, making their clinical use unfeasible. FTIR analysis allowed this investigation. 

  1. How did the bond strength of the adhesives compare between the control group and those with varying concentrations of graphene and graphene modified by L-DOPA?

Thank you for your concern. The bond strength analysis comparing the control group to the other experimental groups was carried out using the one-way ANOVA and Fisher's post hoc test, considering a significance level of 5%. We therefore considered a single variable (dental adhesive) for this analysis, with each concentration of graphene and graphene modified by L-DOPA, and control group as independent groups.

  1. What were the significant conclusions drawn about the properties of graphene and its modification with L-DOPA in the context of dental adhesives?

By modifying graphene with L-DOPA, graphene properties, and therefore the properties of adhesives incorporated by it, were in general enhanced. A total of 0.75 wt% of graphene modified by L-DOPA incorporated in the dental adhesive kept the bond strength and modulus of elasticity similar to the ones of Control group, while improved antibacterial activity, and decreased cytotoxicity, flexural strength, water sorption and solubility.

Reviewer 4 Report

Comments and Suggestions for Authors

The authors have modified graphene nanoparticles by L-dopa in order to improve the properties of experimental dental adhesives. The modification of graphene improved the particles dispersion. Control presented the highest conversion degree, flexural strength and bond strength. This work is interesting.

Specific comments:

1. Please revise Figure 1 and show clear scale bars.

2. Please provide the concentrations for different solutions in the captions for Figure 2.

3. Please provide the stress-strain curves for the samples.

4. The description for the cytotoxicity was too little, please revise it carefully and explain the reason for the higher biocompatibility values than the control.

5. Please revise the abstract and conclusion, and show some data results in them.

Comments on the Quality of English Language

Please carefully check the English grammar.

Author Response

  1. Please revise Figure 1 and show clear scale bars.

Thank you for the suggestion. We have enhanced the scale bars and other details of the figure that maybe were not very visible, including the headings. The figure is now updated in the manuscript (Page 7, Line 326).

  1. Please provide the concentrations for different solutions in the captions for Figure 2.

Thank you very much. The caption of Figure 2 was updated with the concentrations of the solutions (Page 7, Lines 338-339).

  1. Please provide the stress-strain curves for the samples.

We appreciate your suggestion, but unfortunately, we are not able to provide the stress-strain curves for the samples. The 3-point flexural test was performed manually in a universal mechanical test machine, i.e. no software, which could provide the curves, was connected to the machine. The equipment's display showed the value of the force applied to cause the failure of the sample. This value was noted down and entered into the respective formulas to obtain the values for flexural strength and modulus of elasticity.

  1. The description for the cytotoxicity was too little, please revise it carefully and explain the reason for the higher biocompatibility values than the control.

Thank you for pointing this out. Actually, in spite of different numerical values, the biocompatibility of all adhesives was considered statistically similar. As you can check in Figure 8 (Page 12, Line 433), from Control to 0.75% Gd, all adhesives presented letter „b“ above the bars. Similar letters indicate statistical similarity. Negative control was the only one showing 100% of biocompatibility. This result was already expected, because negative control was only fresh culture medium. Following the I.S. EN ISO 10993 and I.S. EN ISO 7405, this group had to be added specifically in this experiment to make sure the culture medium was not somehow contaminated. The fact that negative control presented 100% of biocompatibility assured us that there was no contamination, i.e., the test was accurate. The only groups that presented also statistical similarity to negative control were 0.75% G and Gd. Considering all adhesives showed statistical similarity, but 0.75% G and Gd were also similar to negative control, we focused our discussion of biocompatibility topic on explaining that graphene possibly benefits the growth of fibroblast-like cells. We believe that explaining the reason for those groups to present higher biocompatibility than control would be inappropriate, since the statistical analysis did not show any difference between them. Nevertheless, after revising the text, we do agree that some words we used in this topic of discussion could generate some misunderstanding. Therefore, we have made some changes. The sentence “Analyzing the graph of Figure 8, it is worthy of attention that the increase of G and Gd concentration was proportional to the increase of relative biocompatibility. In fact, the only groups that were statistically similar to negative control were the 0.75% ones.” was replaced to „Analyzing the graph of Figure 8, although all adhesives presented statistical similarity, it is worthy of attention that the only groups that were also statistically similar to negative control (fresh culture medium) were the 0.75% ones.” (Page 13, Lines 555-557).

  1. Please revise the abstract and conclusion, and show some data results in them.

Thank you for the suggestion. We have tried to modify the abstract to include data results in it, but definitely we could not manage limiting the text to 200 words, as requested by the journal, not using multiple abbreviations, so the text would not get confuse for the reader, and adding numerical data for all the results (we understand that adding data for only some of the results would de-standardize the text). We are truly sorry, but this request was not possible to be fulfilled. As for the conclusion, the main results are already somehow summarized in the paragraph (“By modifying graphene with L-DOPA, graphene properties, and therefore the properties of adhesives incorporated by it, were in general enhanced. A total of 0.75 wt% of Gd incorporated in the dental adhesive kept the bond strength and modulus of elasticity similar to the ones of Control group, while improved antibacterial activity, and decreased cytotoxicity, flexural strength, water sorption and solubility.” – Page 14, Lines 601-605). We consider that more detailed results should be kept only in abstract, results or discussion section. Accordingly, the numerical data would be inappropriate in the conclusion.

Response to Comments on the Quality of English Language:

Please carefully check the English grammar.

Thank you very much. We have exerted our best efforts to check and correct any mistakes of English grammar in the whole text, as requested.

Round 2

Reviewer 1 Report

Comments and Suggestions for Authors

Thank you for revising the manuscript. I have no further comments.